# 53-attosecond X-ray pulses reach the carbon K-edge

Jie Li[1], Xiaoming Ren[1], Yanchun Yin[1], Kun Zhao[1,2], Andrew Chew[1], Yan Cheng[1], Eric Cunningham[1], Yang Wang[1], Shuyuan Hu[1], Yi Wu[1], Michael Chini[3] & Zenghu Chang[1,3]

The motion of electrons in the microcosm occurs on a time scale set by the atomic unit of time—24 attoseconds. Attosecond pulses at photon energies corresponding to the fundamental absorption edges of matter, which lie in the soft X-ray regime above 200 eV, permit the probing of electronic excitation, chemical state, and atomic structure. Here we demonstrate a soft X-ray pulse duration of 53 as and single pulse streaking reaching the carbon K-absorption edge (284 eV) by utilizing intense two-cycle driving pulses near 1.8-μm center wavelength. Such pulses permit studies of electron dynamics in live biological samples and next-generation electronic materials such as diamond.

---

[1] Institute for the Frontier of Attosecond Science and Technology, CREOL, University of Central Florida, Orlando, FL 32816, USA. [2] Beijing National Laboratory for Condensed Matter Physics, Institute of Physics, Chinese Academy of Sciences, Beijing 100190, China. [3] Department of Physics, University of Central Florida, Orlando, FL 32816, USA. Correspondence and requests for materials should be addressed to Z.C. (email: zenghu.chang@ucf.edu)

High-energy photon bursts, emitted from a gas ensemble at every half optical cycle within an intense laser electric field, form an attosecond pulse train[1]. Upon restricting these bursts to within one-half cycle of the laser field, a single attosecond pulse may be isolated, demonstrating great importance to the study of electron dynamics in pump-probe experiments[2]. Prior to this study, isolated attosecond pulses (IAPs) were primarily generated using few-cycle near-infrared (NIR) Ti:Sapphire lasers with sub-cycle gating technique, such as amplitude gating[3], polarization gating[4], double optical gating[5], and the attosecond lighthouse[6]. The center photon energy and bandwidth of the generated IAPs are limited to within 100 eV and a few tens of eV, respectively, which has permitted the generation of pulse duration as short as 130 as in 2006[7], later 80 as in ref. [3], and then 67 as in ref. [8], as characterized using photoelectron streaking and phase retrieval techniques[9–11]. These first-generation attosecond light sources offered unprecedented temporal resolution in observing and controlling electron and nuclear dynamics in atoms, molecules, and condensed matter. Recently, attosecond band-gap dynamics were observed in silicon in transient absorption experiments near the Si L-edge at ~ 100 eV[12]. Applying this method to next-generation semiconductor materials such as diamond and graphene requires photon energies surpassing the carbon K-edge (284 eV). Further scaling up of center photon energies and bandwidth, as well as shortening attosecond pulse duration, are technically difficult using 800 nm driving lasers.

The foundation of attosecond light sources is high-order harmonic generation (HHG). It was demonstrated in 2001 that the cutoff of the high harmonic spectrum can be extended dramatically by increasing the driving laser wavelength[13]. The longer period of long-wavelength driving lasers allows an ionized electron more time to be accelerated to a higher kinetic energy in the continuum before recombination with its parent ion, thus leading to higher cutoff photon energies (proportional to the square of the wavelength) when compared to shorter-wavelength NIR drivers. Significant efforts have been made to understand the scaling of the conversion efficiency with longer driving laser wavelengths[14]. Phase matching has also been investigated to overcome the decrease in HHG conversion efficiency due to quantum diffusion effects and the reduction in the recombination cross section[15]. Recent development of carrier-envelope phase (CEP)-stabilized few-cycle lasers at 1.6–2.1 μm paved the way for the next generation of attosecond light sources. CEP-controlled, soft X-ray pulses reaching the water window (284–530 eV) have been generated using these driving lasers[16, 17], and evidence of IAPs therefrom were demonstrated[18–20]. However, no streaking measurement for characterizing the temporal profile of the attosecond pulse in the water window has been reported.

Here we report attosecond streaking measurements of soft X-ray IAPs crossing the boundary of the water window.

Implementing the polarization gating technique, we demonstrate the temporal characterization of the soft X-ray pulses with a record of 53 as.

## Results

**Attosecond Streak Camera**. The experimental set-up is shown in Fig. 1. HHG is driven by a home-built 1 kHz, CEP-stabilized, optical parametric chirped pulse amplifier system with an output of 12 fs (two-cycle) pulses near 1.8 μm[21]. The infrared (IR) beam is split into two arms. The first, high-energy arm (90% of the 1.5 mJ energy) passes through polarization gating optics[20]. The laser beam, now with a time-dependent ellipticity is loosely focused (f = 450 mm) into a 1.5 mm-long neon gas cell to generate isolated soft X-ray pulses[20]. Tin filters are inserted to block the residual IR beam and to compensate the chirp of the attosecond pulses[22]. Using a grazing-incidence, nickel-coated toroidal mirror, the soft X-ray beam is focused through a hole-drilled mirror, which also re-introduces the second, low-energy IR arm (10% of the 1.5 mJ energy) with a variable time delay. The combined soft X-ray and IR pulses are focused onto a helium or neon detection gas jet to generate photoelectrons, which are collected by a 3-meter-long magnetic-bottle electron time-of-flight (TOF) spectrometer[23]. The interference fringes from a 532 nm CW laser co-propagated through both arms are used to feedback control a piezo-electric transducer that controls and stabilizes the time delay between the soft X-ray and the streaking pulses.

**Polarization gating for isolating the attosecond pulses**. Polarization gating has been proven a robust scheme for generating broadband IAPs using two-cycle Ti:Sapphire lasers[24]. It takes advantage of the ellipticity dependence of the HHG yield, which becomes even more effective as the driving laser wavelength increases[20]. The polarization optics in Fig. 1 convert a linearly polarized laser pulse into one within which the field polarization changes from circular to linear and back to circular. Streaking traces are taken for different polarization gate widths (Supplementary Fig. 1) to determine the optimal parameters of the polarization optics. IAPs are observed when the width of the linearly polarized gate is less than half of a laser cycle.

One of the challenges of conducting attosecond streaking near the carbon K-edge is the extremely low photon flux of the soft X-ray emission and the small absorption cross section of the detection gas within the soft X-ray photon energy range. The soft X-ray photon flux should be optimized to produce sufficient photoelectrons to construct streaking traces with a reasonable signal-to-noise ratio. Figure 2 shows the polarization-gated soft X-ray spectra at different gas cell pressures measured by the electron TOF. A generation pressure of 1 bar is chosen as a compromise between signal strength, spectral width, and load on

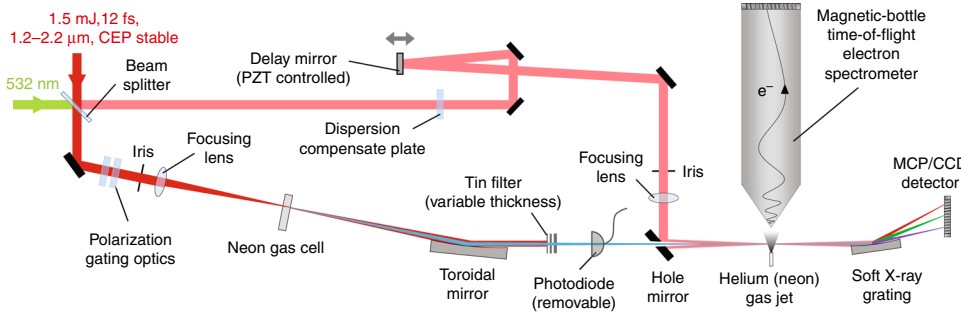

**Fig. 1** Experimental set-up. Schematic illustration for isolated attosecond X-ray pulse generation and characterization. CCD charge-coupled device; MCP microchannel plate, PZT piezo-electric transducer

the turbo pumps. The decrease of cutoff photon energy at high gas pressure is likely caused by ionization-induced plasma defocusing, which decreases the laser intensity at the interaction region and shortens the extension of the plateau harmonics[25].

**53-attosecond pulses retrieved by PROOF method.** The streaked photoelectron spectrum as a function of delay between the soft X-ray pulse and the streaking IR pulse is measured and depicted in Fig. 3a. Helium is used as the detection gas to avoid the contribution of multiple valence orbitals to the photoelectron spectrum. Despite its low absorption cross section for the soft X-ray pulse, the momentum shift of photoelectron is clearly visible across the whole photoelectron spectrum from 100 to 300 eV, indicating the generation of an IAP into the water window.

To characterize such a broadband IAP, the phase retrieval by omega oscillation filtering (PROOF) technique is implemented[11]. In PROOF, the photoelectron spectrogram is broken down into its primary Fourier components:

$$I(\nu,\tau) \approx I_o(\nu) + I_\omega(\nu,\tau) + I_{2\omega}(\nu,\tau) \qquad (1)$$

where $I_o$ does not change with delay $\tau$, $I_\omega$, and $I_{2\omega}$ oscillate with the streaking laser frequency $\omega$ and twice the frequency $2\omega$, respectively, and $\nu$ is the photoelectron momentum. While the soft X-ray spectrum is directly measured by the TOF spectrometer, the unknown spectral phase is encoded in $I_\omega(\nu,\tau)$. During the retrieval, the amplitude and phase of the soft X-ray pulses, depicted in Fig. 3c, d, are guessed iteratively in PROOF to match

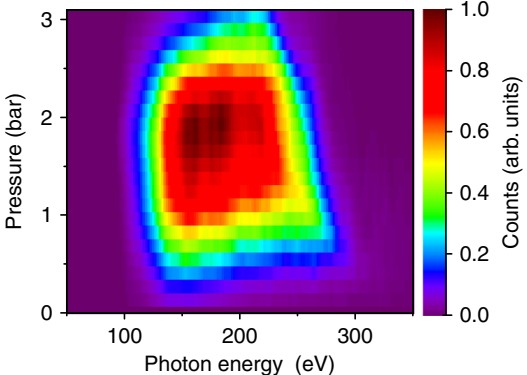

**Fig. 2** Pressure-dependent soft X-ray yield. Soft X-ray continua generated by polarization gating as a function of pressure in the neon gas cell. The spectra were recorded by an electron TOF spectrometer and corrected for the photoionization potential (21.6 eV) and absorption cross section of the neon detection gas. Photons with energy <100 eV were filtered out using a 100 nm tin filter

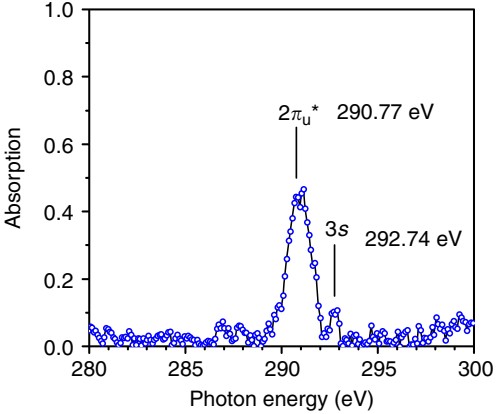

**Fig. 4** Carbon dioxide K-shell photoabsorption spectrum. The two absorption peaks correspond to $C_{1s} \rightarrow 2\pi_u^*$ and $C_{1s} \rightarrow$ Rydberg 3s state[26]. Carbon dioxide gas with 25 torr·mm pressure-length product is used in this measurement

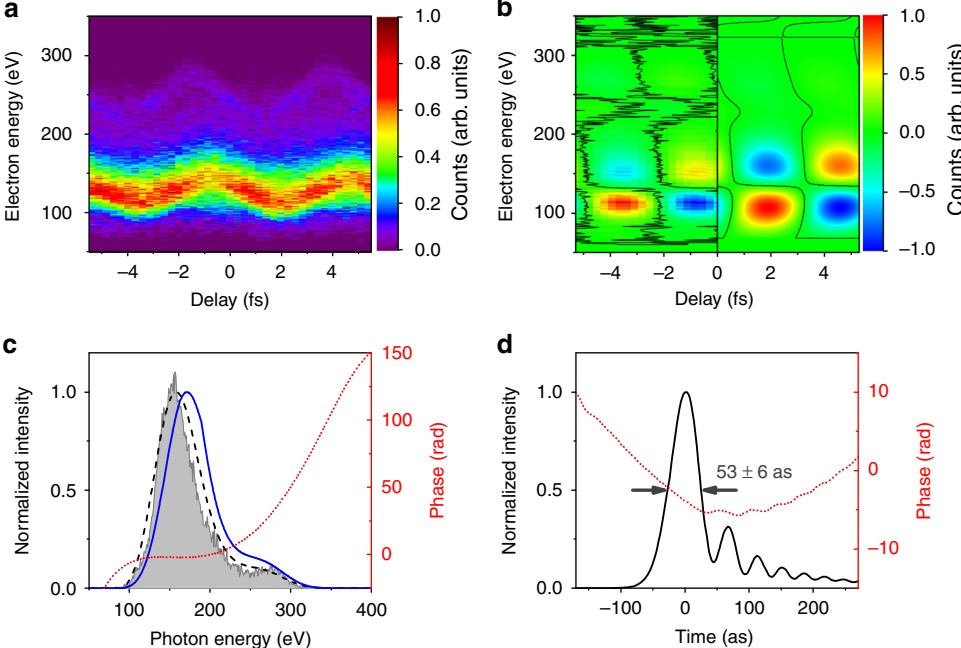

**Fig. 3** PROOF retrieved 53 as soft X-ray pulse. **a** Photoelectron spectrogram as a function of temporal delay between the soft X-ray and the streaking IR pulses in the case of a 400-nm-thick tin filter. A negative delay corresponds to an earlier IR pulse arrival. **b** Filter $I_\omega$ trace (−5 to 0 fs) from the spectrogram in **a** and the retrieved $I_\omega$ trace (0–5 fs). **c** Experimentally recorded (*gray shade*) and PROOF-retrieved spectra (*black dash*) by adding helium photoionization potential (24.6 eV); corrected photon spectrum (*blue solid*), and spectrum phase (*red dot*) from PROOF. **d** Retrieved temporal intensity profile and phase of the 53 as pulses

the retrieved modulation depth and phase angle of $I_\omega$ of the measured trace (Fig. 3b).

The thickness of the tin filter in this measurement is 400 nm. We experimented with different filter thicknesses to demonstrate that a 400-nm-thick tin filter is the best for compensating the atto-chirp under our experimental conditions (Supplementary Fig. 3). However, only the spectral phase error in the low-energy part (<200 eV) of the tin filter transmission window can be well-compensated because the filter's group delay dispersion approaches zero at 300 eV (Supplementary Fig. 2). This is confirmed in Fig. 3a, b, as the up-streaking excursion in the high-energy section above 200 eV is asymmetric horizontally with respect to the turning point of the streaking trace, while the trace below 200 eV is symmetric. The PROOF retrieval also shows a parabolic phase above 200 eV (red dot in Fig. 3c). The retrieved pulse duration full width at half maximum (FWHM) reaches 53 as (transform limited 20 as), which is less than the shortest attosecond pulses achieved with Ti:Sapphire driving lasers. Further increase of the filter thickness leads to an over-compensated chirp for low-energy photons, resulting in a longer pulse duration (Supplementary Fig. 3). The soft X-ray photon flux was measured with an extreme ultraviolet (XUV) photodiode for HHG using the linearly polarized driving field. Knowing the relative intensity change between the ungated and gated HHG, as well as the tin filter transmission, the photon flux for the 53 as pulse is estimated to be $\sim 5 \times 10^6$ photons per laser shot. The photon flux above carbon K-edge (284 eV) is $> 1 \times 10^5$ photons per laser shot.

**K-edge X-ray absorption of carbon dioxide molecule**. To demonstrate the applicability of our water window attosecond source, a carbon K-edge absorption experiment was performed. The gas jet in Fig. 1 was replaced with a 1 mm-long gas cell filled with 25 torr carbon dioxide. A 2400 lines/mm soft X-ray grating and an X-ray CCD camera were chose to achieve high spectrum resolution near carbon K-edge. We integrated for 4 minutes to extract the absorption spectrum shown in Fig. 4. The main absorption peak at 290.77 eV corresponds to the transition from carbon ground state 1s to the lowest unoccupied molecular orbital $2\pi_u^\star$, while the satellite peak at 292.74 eV corresponds to the excitation to the Rydberg 3s orbital[26]. Access to these absorption features allows for the tracing of the temporal evolution of valence π- or σ-bonding electrons in bio-molecules. Such broadband and ultrashort attosecond pulses combined with X-ray absorption near-edge spectroscopy techniques will become powerful tools for studying ultrafast charge dynamics in molecules and condensed-matter targets containing carbon.

Aiming to control chemical reactions via manipulating electron dynamics in molecules with ultrashort laser pulses[27], attosecond water window X-rays will also play an important role in the emerging field of attochemistry. Charge migration, a process strongly affecting chemical reactivity, occurs on sub to few femtosecond time scale[28, 29]. Water window attosecond pulses are unique tools for measuring the charge distribution in molecules containing carbon/oxygen atoms and exploring the possibility to predetermine chemical reaction path by controlling the initial charge migration step[27].

## Discussion

In summary, we have shown the generation and characterization of isolated soft X-ray pulses. With the shortest pulse duration decreased by 20%, the spectral range (100–330 eV) where the high harmonic pulses have been characterized in the time domain is doubled. We also determined experimentally and quantitatively that the attosecond pulse duration is limited by the chirp instead

of the bandwidth in the new spectral region. The bandwidth tunability (10–150 eV) of previously demonstrated IAPs has already proven worthwhile in their applications to the study of inner-shell or valence-electron dynamics. The extremely broad bandwidth (100–330 eV) of the 53 as pulses covers the boron K-edge (188 eV) as well as the carbon K-edge. Very recently, time-resolved X-ray transient absorption in the water window has been demonstrated to observe light-induced chemical reactions in $CF_4$ and other molecules with temporal resolution limited to 40 fs[30]. The X-ray source demonstrated in this work makes it practical to observe charge migration by exploiting transition from the carbon core level to the unoccupied valence orbitals at the carbon K-edge (Fig. 4), similar to that demonstrated in ref. [30] but with attosecond resolution. Such attosecond sources—synchronized with few-cycle IR fields or high-energy XUV pulses—will bring opportunities to study biological or chemical science and strong-field physics.

**Data availability**. The authors declare that the main data supporting the findings of this study are available within the article and its Supplementary Information files. Extra data are available from the corresponding author upon request.

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

## Acknowledgements

We would like to thank Dr. Jens Biegert and his group members from ICFO (Spain) for sharing their experimental data and discussing the results. This work has been supported by the DARPA PULSE program by a grant from AMRDEC (W31P4Q1310017); the Army Research Office (W911NF-14-1-0383, W911NF-15-1- 0336); the Air Force Office of Scientific Research (FA9550-15-1-0037, FA9550-16-1-0149). This material is also based upon work supported by the National Science Foundation under Grant Number Number (NSF Grant Number 1506345). Any opinions, findings, and conclusions or recommendations expressed in this material are those of the authors and do not necessarily reflect the views of the National Science Foundation.

## Author contributions

Z.C. conceived and supervised the study. X.R., Y.Y., J.L., E.C., and Y.W. developed the laser source. J.L. and X.R., (in cooperation with A.C., Y.C., Y.W., and S.H.) prepared and performed the experiment. M.C., K.Z., and J.L. performed simulations and pulse retrieval. J.L. and Z.C. wrote major parts of the manuscript. All authors discussed the results and contributed to the final manuscript.

## Additional information

**Competing interests:** The authors declare no competing financial interests.

**Change History:** A correction to this article has been published and is linked from the HTML version of this article.

