## [Peer review file · Nature Communications]

Reviewers' comments:

Reviewer #1 (Remarks to the Author):

This is a technically impressive paper that demonstrates production of isolated attosecond pulses at high photon energies. The pulses are very short (53 as), and while other groups have claimed to generate attosecond pulses in a similar spectra regime, this paper clearly demonstrates via attosecond streaking that the pulses are indeed isolated attosecond pulses. So as a technical accomplishment the paper is significant. There are, however, a couple of issues that need to be addressed. First, it became apparent only about halfway through the paper that the spectral range of the IAP is mainly below the water window; there appears to be very little intensity above the carbon K-edge. The authors should be a little more forthcoming about this point earlier in the paper. It would also be helpful to estimate the photon flux above 284 eV.

The other aspect of this paper that is a bit strange is that starting on p. 6, a second set of results from a different laser system is presented. There is no information about this system; it is not clear if the only difference is that no double optical gating is used, or if it's a completely different system than that shown in Fig. 1. This part of the paper reads like an add-on. There is no hint of a second system in the abstract, for example. The material here should be better integrated into the rest of the manuscript by, for example, briefly describing the second system as part of the discussion of Fig. 1.

If these issues are addressed, this paper is probably publishable in Nature Communications. The focus of the paper is a bit narrow, covering laser pulse characterization with no application to a model system, but the work is certainly of very high quality.

Reviewer #2 (Remarks to the Author):

The manuscript by Li et al. reports on the production and characterization of isolated attosecond pulses spanning the carbon K-edge around 284 eV. Such a source of soft x-rays is important because of its ability to probe matter using XAFS techniques. Two different laser systems by two teams are employed: UCF (USA) and ICFO (Spain).

The manuscript largely represents a technical achievement. I do not believe that it is appropriate for Nature Communications. Similar studies have been published in Optics Letters.

What is new?

- The generation of an isolated attosecond pulse in the range 130-270 eV (Fig. 2) or 120-220 eV with a tail to 300 eV (Fig. 3c).
- Compensation of the atto chirp below 200 eV using a tin filter.
- Attosecond streaking measurement of the pulse duration at 53 asec.

What has been done previously?

- Popmintchev Science 336, 6086 (2012) – Generation of a spectrum extending to 1.4 keV, composed of a series of attosecond pulses.
- Opt Lett 37, 3893 (2012) [UCF team] – Generation of an isolated 67 asec pulse, 60-150 eV spectrum, using DOG and measured with PROOF.
- Appl Phys Lett 108, 231102 (2016) [UCF team] – same setup as present manuscript, 35-300 eV spectrum. No measurement of pulse duration.
- Opt Lett 18, 5383 (2014) [ICFO team] – Same setup as present manuscript. Spectrum to 400 eV. NEXAFS measurement. Fig 3 same as present Fig. 4a.
- Nat Commun 6, 6611 (2015) [ICFO team] – Same laser setup as present manuscript. Isolated asec pulses with photonic streaking, 230-300 eV spectrum.

In summary, what is new here is the measurement of the pulse duration of 53 asec. All other parts have been done previously at either UCF or ICFO.

Reviewer #3 (Remarks to the Author):

The manuscript by Jie Li et al., entitled as "53-Attosecond X-ray Pulses Glancing Through the Water Window," describes the generation of isolated attosecond pulses in the soft x-ray region, of which the spectrum is extended beyond the carbon K edge, the entrance of the water window (284 - 530 eV).

At UCF, the authors succeed in measuring attosecond spectrograms, which are analyzed by the PROOF method. The spectrograms obtained with Sn filters of different thickness analyzed by the PROOF confirm the effect of filter dispersion and, using a 400-nm-thick Sn filter, the authors confirm the generation of 53-attosecond pulses in the soft x-ray region, which are the shortest light pulses in the world. The results shown in Figs 1 - 3 together with the supplemental information are excellent and worth publishing in Nature Communications.

However, the experimental result (Fig. 4) at ICFO and its interpretation are not convincing. The authors claim the generation of "isolated attosecond" pulses from the streaking spectrogram shown in Fig. 4b. However, the authors do not retrieve the spectrogram to obtain the temporal information of the soft x-ray pulses. Without the pulse duration of the soft x-ray bursts specified, it is inappropriate to use the term "attosecond". Additionally, concerning the term "isolated", the experimental results shown in the manuscript is not enough to claim the isolation of a soft x-ray burst. Please provide CEP dependence of streaking spectrograms and/or high harmonic spectra to estimate the intensity contrast between the primary and satellite pulses.

In summary, the authors are suggested to provide a major revision including additional experimental results to support their claims, modification the interpretation of the experimental results at ICFO, or the removal of the streaking results at ICFO from the manuscript before resubmission for publication.

Response to Reviewer's Comments

We would like to thank all reviewers for their comments, which have been addressed in detail in this response. The manuscript has been revised accordingly.

Reviewer #1 (Remarks to the Author):

Comment 1. This is a technically impressive paper that demonstrates production of isolated attosecond pulses at high photon energies. The pulses are very short (53 as), and while other groups have claimed to generate attosecond pulses in a similar spectra regime, this paper clearly demonstrates via attosecond streaking that the pulses are indeed isolated attosecond pulses. So as a technical accomplishment the paper is significant. There are, however, a couple of issues that need to be addressed. First, it became apparent only about halfway through the paper that the spectral range of the IAP is mainly below the water window; there appears to be very little intensity above the carbon K-edge. The authors should be a little more forthcoming about this point earlier in the paper.

Reply 1: Our photon spectrum peak is indeed below the carbon K-edge. To avoid misleading the readers, we changed the sentence related to spectrum range in the abstract:

“Here we demonstrate a soft X-ray pulse duration of 53 as and single pulse streaking ~~is~~ reaching the “water window” (284 to 530 eV) by utilizing intense two-cycle driving pulses near 1.8-micron center wavelength.”

In the paragraph three, we also revise the first sentence:

“Here we report attosecond streaking measurements of soft X-ray IAPs ~~in the water window~~ crossing the boundary of the water window.”

Comment 2. It would also be helpful to estimate the photon flux above 284 eV.

Reply 2: We estimate the photo flux > 284 eV is 1/40 of the total flux (100-330 eV) base on our spectrum shape, which gives > 1×10^5 photons per laser shot, or > 1×10^8 photons per second above 284 eV. To the best of our knowledge, this flux is still the highest among all reports that claimed to generate high energy pulse within 284-530 eV:

1. [2016 MIT] 1.5×10⁶ photons /s/ 1% bandwidth @350 eV, J. Phys. B: At. Mol. Opt. Phys. 49 155601(2016)
2. [2016 ICFO] 7.3×10⁷ photons /s @ 280-550 eV, Nature Communications 7, 11493 (2016)
3. [2014 University of Tokyo] < 1×10⁴ photons /s @ 284 -350 eV, Nature Communications 5, 3331 (2014)
4. [2014 INRS Canada] 2×10⁷ photons/s @ 280-500 eV, UP.2014.09.Wed.P3.49
5. [2012 Colorado] ~ 10⁷ photons/s @ 284-530 eV, Science 336, 6086 (2012)

It is worth to point out that the phase and duration of the pulses in these five papers were not characterized. We add the estimated photon flux (>284 eV) to our manuscript:

“Knowing the relative intensity change between the ungated and gated HHG, as well as the tin filter transmission, the photon flux for the 53 as pulse is estimated to be $\sim 5 \times 10^6$ photons per laser shot. The photon flux above carbon K-edge (284 eV) is $> 1 \times 10^5$ photons per laser shot.”

Comment 3. The other aspect of this paper that is a bit strange is that starting on p. 6, a second set of results from a different laser system is presented. There is no information about this system; it is not clear if the only difference is that no double optical gating is used, or if it's a completely different system than that shown in Fig. 1. This part of the paper reads like an add-on. There is no hint of a second system in the abstract, for example. The material here should be better integrated into the rest of the manuscript by, for example, briefly describing the second system as part of the discussion of Fig. 1.

Reply 3: We agree with the reviewer that this part reads like an add-on. Similar concerns have been address by reviewer #3. Therefore, we decide to remove the ICFO's experiment from manuscript.

Comment 4. If these issues are addressed, this paper is probably publishable in Nature Communications. The focus of the paper is a bit narrow, covering laser pulse characterization with no application to a model system, but the work is certainly of very high quality.

Reply 4:

To broaden the scope of our paper and demonstrate the potential high impact of the unique x-ray source, we added new results on carbon K-edge absorption experiment with carbon dioxide gas sample. By using a 2400 lines/mm grating and an x-ray CCD camera in the x-ray spectrometer, we reached higher spectrum resolution (~ 0.1 eV per CCD pixel at 284 eV). We were able to identify absorption peaks from K-shell ($1s$) to unoccupied orbitals ($2\pi_u^*$), the potential temporal resolution is in the order of attosecond. This result is added to the main text (Fig. 4). The results laid the foundation for time-resolving chemical bond breaking of carbon containing molecules as wells as for studying a broad range of coupled electron-nuclear dynamics in molecules and condensed matter.

Reviewer #2 (Remarks to the Author):

The manuscript by Li et al. reports on the production and characterization of isolated attosecond pulses spanning the carbon K-edge around 284 eV. Such a source of soft x-rays is important because of its ability to probe matter using XAFS techniques. Two different laser systems by two teams are employed: UCF (USA) and ICFO (Spain).

The manuscript largely represents a technical achievement. I do not believe that it is appropriate for Nature Communications. Similar studies have been published in Optics Letters.

What is new?

- The generation of an isolated attosecond pulse in the range 130-270 eV (Fig. 2) or 120-220 eV with a tail to 300 eV (Fig. 3c).

- Compensation of the atto chirp below 200 eV using a tin filter.
- Attosecond streaking measurement of the pulse duration at 53 asec.

What has been done previously?

- Popmintchev Science 336, 6086 (2012) – Generation of a spectrum extending to 1.4 keV, composed of a series of attosecond pulses.

Reply 1: This laser is at relative low repetition rate (20Hz), limit the photo flux to $\sim 10^7$ photons/s at 284-530 eV. More important, only spectra were measured, which indicate pulse trains. They should not be directly compared with our isolated pulses. Their pulses are likely femtosecond pulses instead of attosecond pulses since their driving laser is at 3.9 micron and there was no chirp compensation.

- Opt Lett 37, 3893 (2012) [UCF team] – Generation of an isolated 67 asec pulse, 60-150 eV spectrum, using DOG and measured with PROOF.

Reply 2: The 800 nm driving field gives limited high-harmonic bandwidth (60-150 eV). Pushing attosecond photon energy into the water window is not simply a repeat of what is done with 800 nm laser. To bring the water-window source to the applicable level (or high enough flux for experiment), many teams are developing new driving laser sources and testing optimal gating methods. We believe our technique achieved the highest photon flux in the water window (see replay 2 for reviewer #1) with the shortest isolated pulse. Which is a milestone in attosecond research.

- Appl Phys Lett 108, 231102 (2016) [UCF team] – same setup as present manuscript, 35-300 eV spectrum. No measurement of pulse duration.

Reply 3: In our APL paper, the CEP effect indicates isolated attosecond pulses are generated. However, the isolated attosecond pulse should only be demonstrated by streaking experiment (include photonic streaking) as reviewer #1 commented. Water window HHG has been demonstrated for many years, but no streaking measurements at 300 eV have ever been achieved until this work. We have solved the long-standing issue of character attosecond pulses by photoelectron streaking at high photon energy range.

- Opt Lett 18, 5383 (2014) [ICFO team] – Same setup as present manuscript. Spectrum to 400 eV. NEXAFS measurement. Fig 3 same as present Fig. 4a.

Reply 4: Attosecond pulse train instead of isolated pulse were produced, should not be compared directly. (Same as reply 1)

- Nat Commun 6, 6611 (2015) [ICFO team] – Same laser setup as present manuscript. Isolated asec pulses with photonic streaking, 230-300 eV spectrum.

Reply 5: This paper is indeed the first demonstration of isolation water-window attosecond pulse via photonic streaking. However, the photon flux is not given.

In summary, what is new here is the measurement of the pulse duration of 53 asec. All other parts have been done previously at either UCF or ICFO.

Reply 6:

Our work has solved the long standing difficulties in generating and characterizing isolated attosecond pulses in the 100---300 eV range. Although the generation (polarization gating) and PROOF techniques have been used for 800 nm driving lasers, whether they could be applied to high energy broadband attosecond pulses was uncertain until our work. We demonstrated that nearly 50 as x-ray pulses covering water window could be generated with photon flux high enough for a photoelectron streaking measurement (for the first time). Excellent signal to noise ratio was achieved at the K-edge of carbon for gas samples (figure 4 add in the main text), which paved the road for investigating chemical reaction dynamics using the timed resolved X-ray Absorption Near Edge Spectroscopy (XANES).

Reviewer #3 (Remarks to the Author):

The manuscript by Jie Li et al., entitled as “53-Attosecond X-ray Pulses Glancing Through the Water Window,” describes the generation of isolated attosecond pulses in the soft x-ray region, of which the spectrum is extended beyond the carbon K edge, the entrance of the water window (284 - 530 eV). At UCF, the authors succeed in measuring attosecond spectrograms, which are analyzed by the PROOF method. The spectrograms obtained with Sn filters of different thickness analyzed by the PROOF confirm the effect of filter dispersion and, using a 400-nm-thick Sn filter, the authors confirm the generation of 53-attosecond pulses in the soft x-ray region, which are the shortest light pulses in the world. The results shown in Figs 1 - 3 together with the supplemental information are excellent and worth publishing in Nature Communications.

However, the experimental result (Fig. 4) at ICFO and its interpretation are not convincing. The authors claim the generation of “isolated attosecond” pulses from the streaking spectrogram shown in Fig. 4b. However, the authors do not retrieve the spectrogram to obtain the temporal information of the soft x-ray pulses. Without the pulse duration of the soft x-ray bursts specified, it is inappropriate to use the term “attosecond”. Additionally, concerning the term “isolated”, the experimental results shown in the manuscript is not enough to claim the isolation of a soft x-ray burst. Please provide CEP dependence of streaking spectrograms and/or high harmonic spectra to estimate the intensity contrast between the primary and satellite pulses.

In summary, the authors are suggested to provide a major revision including additional experimental results to support their claims, modification the interpretation of the experimental results at ICFO, or the removal of the streaking results at ICFO from the manuscript before resubmission for publication.

Reply: We agree with the reviewer and removed the experiment results of ICFO.

==== END OF COMMENTS =====

REVIEWERS' COMMENTS:

Reviewer #3 (Remarks to the Author, based on the previous comments of Reviewer #1 and the authors' previous response):

Overall, the responses from the authors on the comments 1 - 4 raised by the Reviewer 1 are fine enough so that they would convince the Reviewer 1. The comments 1, 3, and 4 deal with non-technical issues and are responded well. The comment 2 about the photon flux is a technical issue. Although I cannot guarantee validity of the estimated photon flux because a measurement procedure is not well described, the photon flux is not an essential part of the work and the Reviewer 1 also indicated so in the comment 2 as "It would also be helpful to estimate the photon flux above 284 eV." I would guess the photon flux of their beamline is quite high as they claimed, because they are able to achieve time-resolved soft-x-ray photoelectron spectroscopy, detection efficiency of which is usually lower than direct photon detection. Time-resolved measurement has been highly demanded in the ultrafast community and their achievement is impressive.

Please find below my concerns on their policy about the references. In the manuscript's reference, it would be fair to include some of the five references mentioned in the reply 2. Among the five works, only the second reference, "[2016 ICFO] 7.3×10^7 photons /s @ 280-550 eV, Nature Communications 7, 11493 (2016)," which is self-reference for the group, is included. These works are closely related to the present work, but not referenced well.

Reviewer #2 (Remarks to the Author):

The major change made to the manuscript was to remove the material from ICFO. Although I feel sorry for the ICFO team, I think that this has made the manuscript more coherent. Also new to the revised manuscript is the demonstration of carbon K-shell photoabsorption at 290 eV. This demonstrates that the attosecond source has sufficient flux at the carbon K-edge for spectroscopy experiments.

The manuscript continues to report mostly technological improvements on attosecond sources, which would be more appropriate for Optics Letters. However the other referees seem to lean towards acceptance by Nature Communications, and I will not block it.

(1) The claim of "water window" still seems a stretch, given that the water window is 280- 540 eV. The spectra shown in Fig. 3 peak at 180 eV, with a tail barely reaching the carbon K-edge. Perhaps the claim could be "reaching the carbon K-edge".

(2) The conclusion now includes a suggestion that a 50 attosecond source would give better time resolution for transient absorption experiments. I am not sure about this claim. Since transient absorption involves the creation of an electronic coherence that persists for a time much longer than the excitation pulse, it is not clear how a 50 asec time resolution will be obtained. For example, the absorption line widths shown in Fig. 4 correspond to an emission time in the femtosecond range.

(3) The pulse duration of the spectral portion above 250 eV is unlikely to be near 50 asec, given the measured spectral phase.

Reviewer #3 (Remarks to the Author):

The manuscript by Jie Li et al., entitled as "53-Attosecond X-ray Pulses Glancing Through the Water Window," describes the generation of isolated attosecond pulses in the soft x-ray region.

The authors succeed in retrieving isolated attosecond pulses from measured attosecond spectrograms using the PROOF method. The spectral range of the photoelectron spectra reach the carbon K edge at 284 eV, the entrance of the water window (284 - 530 eV), for the first time. The spectrograms obtained with Sn filters with the different thicknesses confirm the effect of filter dispersion and, using a 400-nm-thick Sn filter, the authors realize the measurement of 53-attosecond pulses, which are the shortest light pulses in the world. The results shown in Figs 1 - 3 together with the supplemental information are convincing, novel and excellent.

An absorption measurement is added in Fig.4, which is a currently hot topics in ultrafast laser community when it is combined with time-resolved measurement. In this context, the authors may add an article "Femtosecond x-ray spectroscopy of an electrocyclic ring-opening reaction," by A. R. Attar from Science as a reference.

I believe this manuscript is worth publishing in Nature Communications.

Response to Reviewer's Comments

We would like to thank Reviewers #2 and #3 again for their comments, which have been addressed in detail in this response. The manuscript has been revised accordingly.

Reviewer #1 (Reviewer #3's comments of Reviewer #1 and our responses from previous round)

Overall, the responses from the authors on the comments 1 - 4 raised by the Reviewer 1 are fine enough so that they would convince the Reviewer 1. The comments 1, 3, and 4 deal with non-technical issues and are responded well. The comment 2 about the photon flux is a technical issue. Although I cannot guarantee validity of the estimated photon flux because a measurement procedure is not well described, the photon flux is not an essential part of the work and the Reviewer 1 also indicated so in the comment 2 as "It would also be helpful to estimate the photon flux above 284 eV." I would guess the photon flux of their beamline is quite high as they claimed, because they are able to achieve time-resolved soft-x-ray photoelectron spectroscopy, detection efficiency of which is usually lower than direct photon detection. Time-resolved measurement has been highly demanded in the ultrafast community and their achievement is impressive.

Please find below my concerns on their policy about the references. In the manuscript's reference, it would be fair to include some of the five references mentioned in the reply 2. Among the five works, only the second reference, "[2016 ICFO] 7.3×10⁷ photons /s @280-550 eV, Nature Communications 7, 11493 (2016)," which is self-reference for the group, is included. These works are closely related to the present work, but not referenced well.

Reply: we agree with the Reviewer and add the works "*Carrier-envelope phase-dependent high harmonic generation in the water window using few-cycle infrared pulses. Nat. Commun. 5, 3331 (2014).*" and "*Water-window soft x-ray high-harmonic generation up to the nitrogen K-edge driven by a kHz, 2.1 μm OPCPA source. J. Phys. B: At. Mol. Phys. 49, 155601 (2016)*" as reference # 16 & 17 in second paragraph of the manuscript:

"Recent development of carrier-envelope phase-stabilized few-cycle lasers at 1.6 to 2.1 μm paved the way for the next generation of attosecond light sources. Carrier-envelope phase (CEP)-controlled, soft X-ray pulses reaching the water-window (284-530 eV) have been generated using these driving lasers [16, 17], and evidence of IAPs therefrom were demonstrated [18-20]."

Reviewer #2 (Remarks to the Author):

The major change made to the manuscript was to remove the material from ICFO. Although I feel sorry for the ICFO team, I think that this has made the manuscript more coherent. Also new to the revised manuscript is the demonstration of carbon K-shell photoabsorption at 290 eV. This demonstrates that the attosecond source has sufficient flux at the carbon K-edge for spectroscopy experiments.

The manuscript continues to report mostly technological improvements on attosecond sources, which would be more appropriate for Optics Letters. However the other referees seem to lean towards acceptance by Nature Communications, and I will not block it.

(1) The claim of "water window" still seems a stretch, given that the water window is 280- 540 eV. The spectra shown in Fig. 3 peak at 180 eV, with a tail barely reaching the carbon K-edge. Perhaps the claim could be "reaching the carbon K-edge".

Reply 1: we agree with the reviewer and change the title to “53-Attosecond X-ray Pulses Reach the Carbon K-edge”. The word “water window” is also change to “carbon K-absorption edge” in the abstract:

“Here we demonstrate a soft X-ray pulse duration of 53 as and single pulse streaking reaching the carbon K-absorption edge (284 eV) by utilizing intense two-cycle driving pulses near 1.8- μ m center wavelength.”

(2) The conclusion now includes a suggestion that a 50 attosecond source would give better time resolution for transient absorption experiments. I am not sure about this claim. Since transient absorption involves the creation of an electronic coherence that persists for a time much longer than the excitation pulse, it is not clear how a 50 asec time resolution will be obtained. For example, the absorption line widths shown in Fig. 4 correspond to an emission time in the femtosecond range.

Reply 2: The absorption line width in Fig. 4 is determined by electron state lifetime, which is indeed in femtosecond range. However, in an attosecond transient absorption experiment, we are interested in the changes of such absorption line, such as the changes in its width, position and intensity, which only depend on the time scale of the physical processes that are initiated by an additional pump laser. Therefore, the time resolution of such measurement can be as high as 50 attosecond using our source. Furthermore, X-ray pulses may be absorbed by a group of states in transient absorption that leads to the formation of electronic wavepackets. The energy range of these states may be so large that 50 as pulses are needed to cover them and assure the synchronization of their excitations.

(3) The pulse duration of the spectral portion above 250 eV is unlikely to be near 50 asec, given the measured spectral phase.

Reply 3: As we already stated in the original manuscript, “only the spectral phase error in the low-energy part (<200 eV) of the tin filter transmission window can be well-compensated because the filter’s group delay dispersion approaches zero at 300 eV”. The achieved 50 asec pulses consist the entire bandwidth and spectral phase. If the spectral phase above 300 eV can be further compensated, a pulse duration less than 50 asec can be achieved.

Reviewer #3 (Remarks to the Author):

The manuscript by Jie Li et al., entitled as “53-Attosecond X-ray Pulses Glancing Through the Water Window,” describes the generation of isolated attosecond pulses in the soft x-ray region. The authors succeed in retrieving isolated attosecond pulses from measured attosecond spectrograms using the PROOF method. The spectral range of the photoelectron spectra reach the carbon K edge at 284 eV, the entrance of the water window (284 - 530 eV), for the first time. The spectrograms obtained with Sn filters with the different thicknesses confirm the effect of filter dispersion and, using a 400-nm-thick Sn filter, the authors realize the measurement of 53-attosecond pulses, which are the shortest light pulses in the world. The results shown in Figs 1 - 3 together with the supplemental information are convincing, novel and excellent.

An absorption measurement is added in Fig.4, which is a currently hot topics in ultrafast laser community when it is combined with time-resolved measurement. In this context, the authors may add an article “Femtosecond x-ray spectroscopy of an electrocyclic ring-opening reaction,” by A. R. Attar from Science as a reference.

I believe this manuscript is worth publishing in Nature Communications.

Reply: we added the work “Femtosecond x-ray spectroscopy of an electrocyclic ring-opening reaction” as reference # 29 after Fig.4.

“Charge migration, a process strongly affecting chemical reactivity, occurs on sub to few femtosecond time scale [28, 29].”